# Disaggregating Asian American and Pacific Islander Risk of Fatal Police Violence

Gabriel L. Schwartz[1]*, Jaquelyn L. Jahn[2]

1 Philip R. Lee Institute for Health Policy Studies, University of California, San Francisco, San Francisco, CA, United States of America, 2 The Ubuntu Center on Racism, Global Movements, and Population Health Equity and the Department of Epidemiology & Biostatistics, Drexel University Dornsife School of Public Health, Philadelphia, PA, United States of America

* gabelschwartz@gmail.com

**Data Availability Statement:** To facilitate future work, numerator data is freely available for download via the Harvard Dataverse (https://doi.org/10.7910/DVN/ZNBVKZ). Denominator data is publicly available via the US Census.

## Abstract

High rates and racial inequities in U.S. fatal police violence are an urgent area of public health concern and policy attention. Asian Americans and Pacific Islanders (AAPIs) have been described as experiencing low rates of fatal police violence, yet AAPI subgroups vary widely on nearly every demographic and economic metric. Here, we calculate fatal police violence rates by AAPI regional and national/ethnic background, finding wide variation. We compile a list of AAPI people killed in interactions with police in 2013–2019, then use web searches and surname algorithms to identify decedents' backgrounds. Rates are then calculated by combining this numerator data with population denominators from the American Community Survey and fitting Poisson models. Excluding 18% of deaths with missing regional backgrounds, East and South Asian Americans died at a rate of 0.05 and 0.04 deaths per 100,000 (95% CI: 0.04–0.06 and 0.02–0.08), respectively, less than a third of Southeast Asian Americans' rate (0.16, CI: 0.13–0.19). Pacific Islanders suffered higher rates (0.88, CI: 0.65–1.19), on par with Native and Black Americans. More granularly, Southeast Asian American groups displaced by US war in Southeast Asia suffered higher rates than others from the same region. Traditional racial classifications thus obscure high risks of fatal police violence for AAPI subgroups. Disaggregation is needed to improve responses to fatal police violence and its racial/ethnic inequities.

## Introduction

Recent advances in the US's monitoring of fatal police violence, defined here as fatalities in police custody or involving the police that would not have occurred in the absence of police intervention, have enabled a more accurate accounting of these deaths, especially with respect to racial inequities [1,2]. Previous work has found high rates among Black and American Indian/Alaskan Native, and lower rates for white and Asian, people [3]. Yet important gaps in describing rates and inequities remain: no previous research has estimated rates of fatal police violence for disaggregated Asian American and Pacific Islander (AAPI) ethnic groups.

**Funding:** The authors received no specific funding for this work.

**Competing interests:** The authors have declared that no competing interests exist.

Studies examining the diversity of experiences with, and perceptions of, police across AAPI ethnic groups are scarce [4,5]. Indeed, federal data on arrests, police contact, and deaths in police custody do not include ethnicity information, except for Hispanic and Pacific Islander classifications. National violence monitoring data, such as the National Violent Death Reporting System, also lack Asian ethnicity information.

It is expected that a disaggregated examination of fatal police violence within those racially classified as Asian would reveal ethnic differences because of unequal histories of colonization, immigration, and conflict that play a role in socioeconomic position and police contact in the US [5–7]. Indeed, income inequality is wider within AAPIs than within other racial groups, and has increased 77% since 1970—reflecting patterns of refugee immigration and federal policy prioritization of "high-skill" immigrants [8,9]. Access to adequate and culturally appropriate mental health services is also not equal across Asian ethnic groups [10,11], which may contribute to inequities in mental health crises that end in fatal police violence.

This analysis therefore identified the ethnicity of AAPI decedents in the most comprehensive databases of fatal police violence in the U.S., Fatal Encounters and Mapping Police Violence. We then estimated average annual rates for AAPI subgroups (2013–2019) using Poisson models.

## Methods

Our analysis proceeded as follows. First, we compiled a list of AAPI decedents using the two most comprehensive databases of these deaths in the U.S. Second, we "hand-coded" each decedent by using public documentation (e.g., newspaper articles, obituaries, etc.) to determine their national and regional backgrounds, in some cases using this hand-coding to correct people erroneously coded as AAPI or as some other race/ethnicity in the databases from which we first compiled cases. Third, we applied imputation algorithms to fill in missing national or regional background codes. And fourth, we fit models to estimate rates for each group. We describe each below.

### Data on fatal police violence

We used data on police-related deaths from Fatal Encounters (FE) (2013–2019, downloaded 10/7/2020), which identifies cases in real time using media reports and public records. FE's methodology yields more comprehensive counts than official vital statistics [12] and federally collected data, which underestimate police-related fatalities during these years [13]. Fatal Encounters started tracking police-involved fatalities in real time in 2013; it also began retrospective searches for these fatalities going back as far as 2000. Because widespread media coverage of these deaths was less common before the early 2010s (when the Black Lives Matter movement came to prominence), the accuracy of retrospective record-keeping and race classification within Fatal Encounters is likely to be lower before 2013. We thus subset to fatalities occurring in 2013 through 2019, the last complete year of data available at the time of our download. We further conservatively restricted to those who were lethally shot, tasered, asphyxiated, beaten, or (in two cases) whose cause of death was unknown, broadly excluding causes of death that could be considered "accidents" or that would also have occurred in the absence of police intervention (e.g., falling from a height, vehicle collisions) [2,3].

Racial classification in FE uses six categories: American Indian/Alaska Native, Asian/Pacific Islander, Black, White, Hispanic, and Middle Eastern. To ensure we were not missing AAPI cases or misidentifying the race/ethnicity of AAPI decedents, we cross-referenced with Mapping Police Violence (MPV), a similar and overlapping database. No decedents were identified by MPV but not by FE. In cases where MPV identified a decedent as AAPI but FE identified

them as some other race/ethnicity (or was missing race/ethnicity data), we performed additional "hand-coding" searches to see if we could find publicly available evidence corroborating MPV's coding (i.e., evidence that a person was in fact AAPI). Of these 13 cases, we were able to corroborate MPV's coding in 3 cases (which were subsequently added to our list of API decedents), conclusively rejected MPV's coding in 4 cases (not added to our list), and were unable to corroborate nor reject MPV's coding in 6 cases. In the latter 6 cases, we defaulted to FE's coding. We also researched those classified as "Middle Eastern" by FE to identify misclassified South Asians; otherwise, "Middle Easterners" were coded as White in our analysis, as they are often listed as such by the Census [14] and our analysis required Census denominators (which we used as offsets in our Poisson models; see below). In total, this yielded 167 AAPI decedents (see below). We then merged these records with 5-year estimates for 2013–2018 from the American Community Survey, which provided denominators for available subgroups. Finally, we created denominators for people with backgrounds from four API regions (Pacific Islanders; East, Southeast, and South Asian) by summing national/ethnic/racial ACS categories, noting that there were some ethnicities with no specific corresponding population denominator data (see S2 Table in S1 Appendix).

Importantly, though rare, Fatal Encounters can include deaths in police custody that were in fact killings performed by other people during a confrontation with police. (For example, take the specific case in our initial list where Person A had a gun and was holding Person B hostage, then police killed Person A; but Person B appeared to have been killed by Person A either during or prior to the police engaging Person A. FE would include both Person A and Person B in their database.) Since, through our hand-coding, we read extensive news coverage on every AAPI death included in FE, we were able to remove any cases equivalent to Person B. We found 4 such cases, who were not included in our final analytic list.

## Asian national/ethnic and regional background identification

We identified decedents' national/ethnic and regional backgrounds in four steps. First, we used FE documentation and web searches to find obituaries, news reports, or public records with ethnicity information. We thus "hand-coded" 81 decedents (49%), using identified national/ethnic background to determine regional background. Second, we used MPV race/ethnicity (which treats Asian and Pacific Islander as separate categories) to impute "Pacific Islander" as decedents' regional background in cases where hand-codes were missing, identifying 24 decedents (14%). Third, we used a previously-validated [15] surname list generated by Lauderdale [16] to identify individuals as Chinese, Filipino, Japanese, Vietnamese, Korean, or Indian, imputing backgrounds for an additional 26 decedents (16%). Fourth, we used NamePrism, an algorithm that predicts ethnic classifications using surnames, yielding national/ethnic backgrounds for 5 and regional backgrounds for 6 additional decedents (3.4%) [17]. In total, we were able to identify national/ethnic background for 112 (67%) and regional background for 137 (82%) decedents. A breakdown of classification method across ethnic groups, showing how many decedents in each group were identified using each imputation method, is available in the Appendix (S1 Table in S1 Appendix).

To assess the accuracy of our imputations, we compared backgrounds as predicted by each algorithm (Lauderdale, NamePrism) to backgrounds as determined via hand-codes. For NamePrism, the algorithm provided a probability score for each predicted ethnicity classification (e.g., the algorithm could predict that, given its list of surnames, a given individual had a 70% chance of being Chinese, a 20% chance of being Vietnamese, and a 10% chance of being Lao). We only used NamePrism's ethnicity predictions for those with a single ethnicity that scored at least 50. Using this cutoff, NamePrism correctly identified 84% of hand-coded

national/ethnic backgrounds and 92% of hand-coded regional backgrounds (see S2 Fig in S1 Appendix). For Lauderdale, there was 66% agreement with our hand-coded national/ethnic background IDs, suggesting reasonable fidelity.

After these procedures, there were 30 decedents for whom we were not able to identify regional background, and 55 decedents for whom we were not able to identify national/ethnic background. In both cases, but especially for national/ethnic background, we expect disproportionate missingness for deaths among Native Hawaiians. This is because we were only able to "hand-code" national/ethnic background for 15% of the 34 deaths in Hawai'i, compared to 58% of decedents in other states. Hawaiian news media rarely reported on the specific national/ethnic background of decedents, potentially because of the high prevalence of multi-racial/multiethnic identities among Hawaiian residents [18]. Further, neither imputation algorithm included Native Hawaiian as a detectable ethnicity.

Finally, to assess whether alternative imputation schemes would change our results, we re-imputed our results, this time beginning with NamePrism and only including imputations from Lauderdale where NamePrism codes were missing. We report the impact of these checks as a sensitivity analysis.

## Statistical analysis

To calculate average annual rates of death by police violence for people of different racial, and AAPI national/ethnic and regional, backgrounds, we fit Poisson models predicting total deaths in a given group in a given year, with fixed effects for each group and corresponding population offsets (representing the total population of each group, which effectively serve as our denominators for calculating fatality rates). These consisted of a single model predicting all national/ethnic group rates, and a second model predicting all regional rates (as well as rates for other racial/ethnic groups, for comparison). These were fit on "group-year" files, i.e., files in which each national/ethnic group was represented by a series of rows, each representing a different year (a "long" longitudinal file), or in which each regional background was represented by a series of such rows. Associated 95% "confidence intervals" should not be interpreted as being centered on realized rates (since we are using population denominators and our numerators are undercounted–true rates were higher than our estimates), but rather confidence intervals around the (under)estimated underlying population risk that gave rise to these realized rates.

Standard errors were calculated using the Huber-White/"sandwich" estimator. Imputation, data organization, and visualization was completed using the open source statistical software R (version 4.0.4); models were run using the statistical software StataMP (version 16).

## Ethical approval

Because the information we analyzed was (A) public, (B) not about living individuals (our data was on decedents), and (C) not collected via intervention or interaction with individuals, this study did not meet the definition of Human Subjects Research. IRB review was therefore not required.

## Results

Of the 137 AAPI decedents with identified regional backgrounds, 65 were of Southeast Asian, 37 of Pacific Islander, 21 of East Asian, and 14 of South Asian descent (Table 1). In raw counts, three ethnonational backgrounds were most prevalent, with 28 Vietnamese, 17 Filipino, and 16 Chinese decedents; no other AAPI ethnic/national group had more than 7 identified deaths, though we note that the ethnic/national background of 55 decedents could not be identified

**Table 1. Characteristics of police violence fatalities, 2013–2019.**

| Characteristic | Asian & Pacific Islander (n = 167) | American Indian & Alaska Native (n = 46) | Black (n = 822) | Hispanic (n = 500) | White (n = 1620) |
|---|---|---|---|---|---|
| **Asian Ethnicity (Region)** | | N/A | N/A | N/A | N/A |
| East | 21 (12.6%) | | | | |
| Pacific Islander | 37 (22.2%) | | | | |
| South | 14 (8.4%) | | | | |
| Southeast | 65 (38.9%) | | | | |
| Missing | 30 (18.0%) | | | | |
| **Age** | | | | | |
| Mean (SD) | 47.4 (11.9) | 32.0 (9.9) | 31.9 (11.2) | 33.4 (10.9) | 39.4 (13.6) |
| Missing | 1 (0.6%) | 1 (2.2%) | 5 (0.6%) | 4 (0.8%) | 5 (0.3%) |
| **Gender** | | | | | |
| Female | 10 (6.0%) | 5 (10.9%) | 33 (4.0%) | 8 (1.6%) | 97 (6.0%) |
| Male | 157 (94.0%) | 41 (89.1%) | 788 (95.9%) | 490 (98.0%) | 1522 (94.0%) |
| Transgender | 0 | 1 (2.2%) | 1 (0.1%) | 2 (0.4%) | 1 (0.1%) |
| **Cause of Death** | | | | | |
| Asphyxiated/Restrained | 3 (1.8%) | 0 | 11 (1.3%) | 6 (1.2%) | 15 (0.9%) |
| Beaten | 2 (1.2%) | 0 | 6 (0.7%) | 4 (0.8%) | 9 (0.6%) |
| Gunshot | 156 (93.4%) | 44 (95.7%) | 756 (92.0%) | 468 (93.6%) | 1546 (95.4%) |
| Tasered | 4 (2.4%) | 1 (2.2%) | 44 (5.4%) | 20 (4.0%) | 45 (2.8%) |
| Undetermined | 2 (1.2%) | 1 (2.2%) | 5 (0.6%) | 2 (0.4%) | 5 (0.3%) |
| **Census Region** | | | | | |
| Midwest | 18 (10.8%) | 9 (19.6%) | 166 (20.2%) | 32 (6.4%) | 303 (18.7%) |
| Northeast | 5 (3.0%) | 0 | 94 (11.4%) | 22 (4.4%) | 115 (7.1%) |
| South | 38 (22.8%) | 8 (17.4%) | 445 (54.1%) | 148 (29.6%) | 718 (44.3%) |
| West | 106 (63.5%) | 29 (63.0%) | 117 (14.2%) | 298 (59.6%) | 484 (29.9%) |

Note: The sample of decedents was restricted to those who were lethally shot, tasered, asphyxiated, beaten, or whose cause of death was unknown, broadly excluding causes of death that could be considered "accidents" or that may also have occurred in the absence of police intervention (e.g., falling from a height, vehicle collisions).

(S3 Table in S1 Appendix). AAPI decedents were overwhelmingly men, on average older than decedents of other racial/ethnic minority groups, and concentrated in the Western US (63.5%) (Table 1).

Goodness-of-fit statistics indicated regular Poisson models were a reasonable fit to these data (for national/ethnic background: deviance goodness-of-fit $\chi^2$: p = 0.930, Pearson goodness-of-fit $\chi^2$: p = 0.533, Wald test of improved fit over a null model: p<0.0001; for regional background, deviance goodness-of-fit $\chi^2$: p = 0.656, Pearson goodness-of-fit $\chi^2$: p = 0.747, Wald test of improved fit over a null model: p<0.0001). Estimates calculated via negative binomial models were functionally identical (and $\alpha$ values assessing overdispersion were effectively 0 for both models).

Annual rates of fatal police violence varied widely by regional background. Rates for East and South Asian Americans were 0.05 and 0.04 deaths per 100,000 (95% CI: 0.04–0.06 and 0.02–0.08), respectively, less than a third the rate for Southeast Asian Americans (0.16, CI: 0.13–0.19). Rates for Pacific Islanders were much higher, on par with those suffered by American Indian/Alaskan Native and Black Americans (0.88, CI: 0.65–1.19) (Fig 1).

Rates similarly varied by national/ethnic background within regions (Fig 2; for a full table of predicted rates and confidence intervals, see S4 Table in S1 Appendix). Among Southeast Asian Americans, some rates were near zero, while national/ethnic groups displaced by US war in Southeast Asia (Vietnamese, Cambodian, Lao, and Hmong) had rates between 0.22 and 0.35, although confidence intervals for these estimates were wide and sometimes overlapped with those of other national/ethnic groups. National/ethnic rates for the Federated States of Micronesia (n = 3 decedents) were not estimated, as they lacked a corresponding Census denominator.

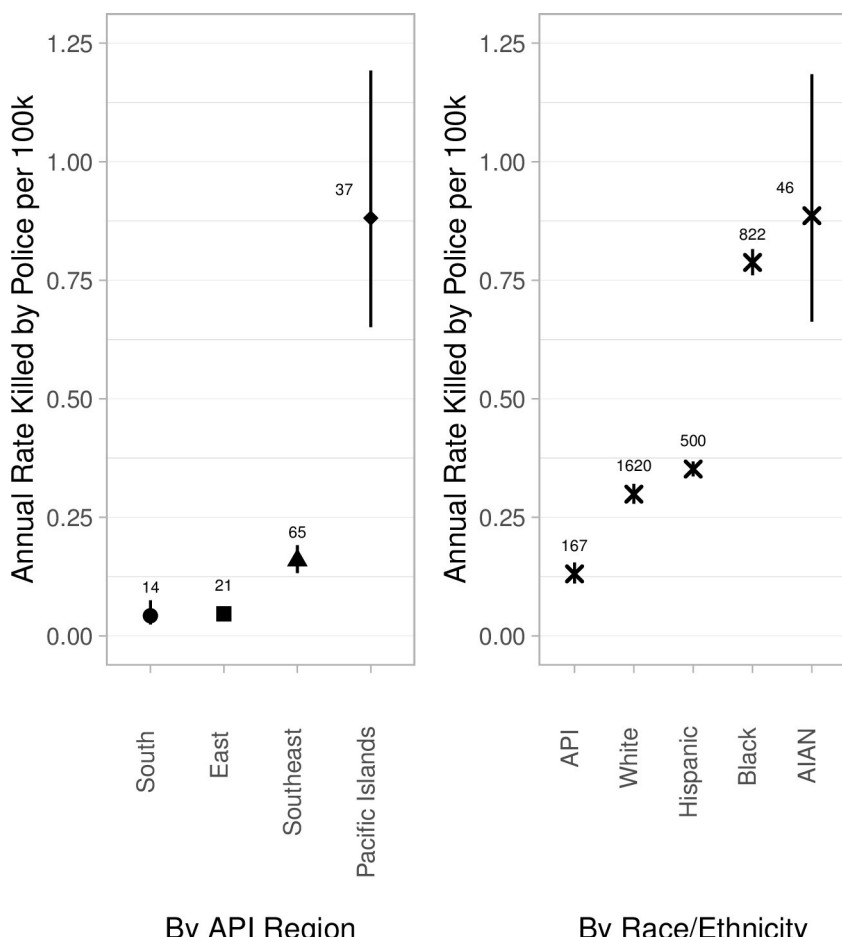

By API Region By Race/Ethnicity

**Fig 1. Estimated annual rates of fatal police violence by AAPI regional background and racial/ethnic group, 2013–2019.** Note: Vertical bars represent 95% confidence intervals. Numbers hovering above point estimates represent the number of decedents informing the estimation of each rate, calculated using a single Poisson model. Total numerator N = 137 deaths; 30/167 additional API decedents were missing regional background data. Note that there were 24 Pacific Islander and 1 Southeast Asian decedents included in Fig 1 for whom we were not able to identify national/ethnic background and are thus not represented in Fig 2. Denominators by race/ethnicity (the right panel) are from the US Census; racial/ethnic groups are treated as mutually exclusive (i.e., the "White" group represents non-Hispanic White people, etc.).

## Sensitivity analyses

To check how changes in the order of our imputation strategy may have affected our results, we reversed the order in which we applied our imputation algorithms: instead of applying Lauderdale and then applying NamePrism to cases that remained un-imputed, we did the reverse. This changed our coding of 3 cases total out of 167, yielding one additional Chinese case, one additional Filipino case, and one fewer Korean case. All of these were among our groups with the lowest number of deaths and largest denominators, meaning these changes made little difference to our estimates, nor did they change any of our conclusions.

## Discussion

We identified large variation in the rates at which different AAPI groups experience fatal police violence. Rates were particularly high for Pacific Islanders. These dramatic differences underscore

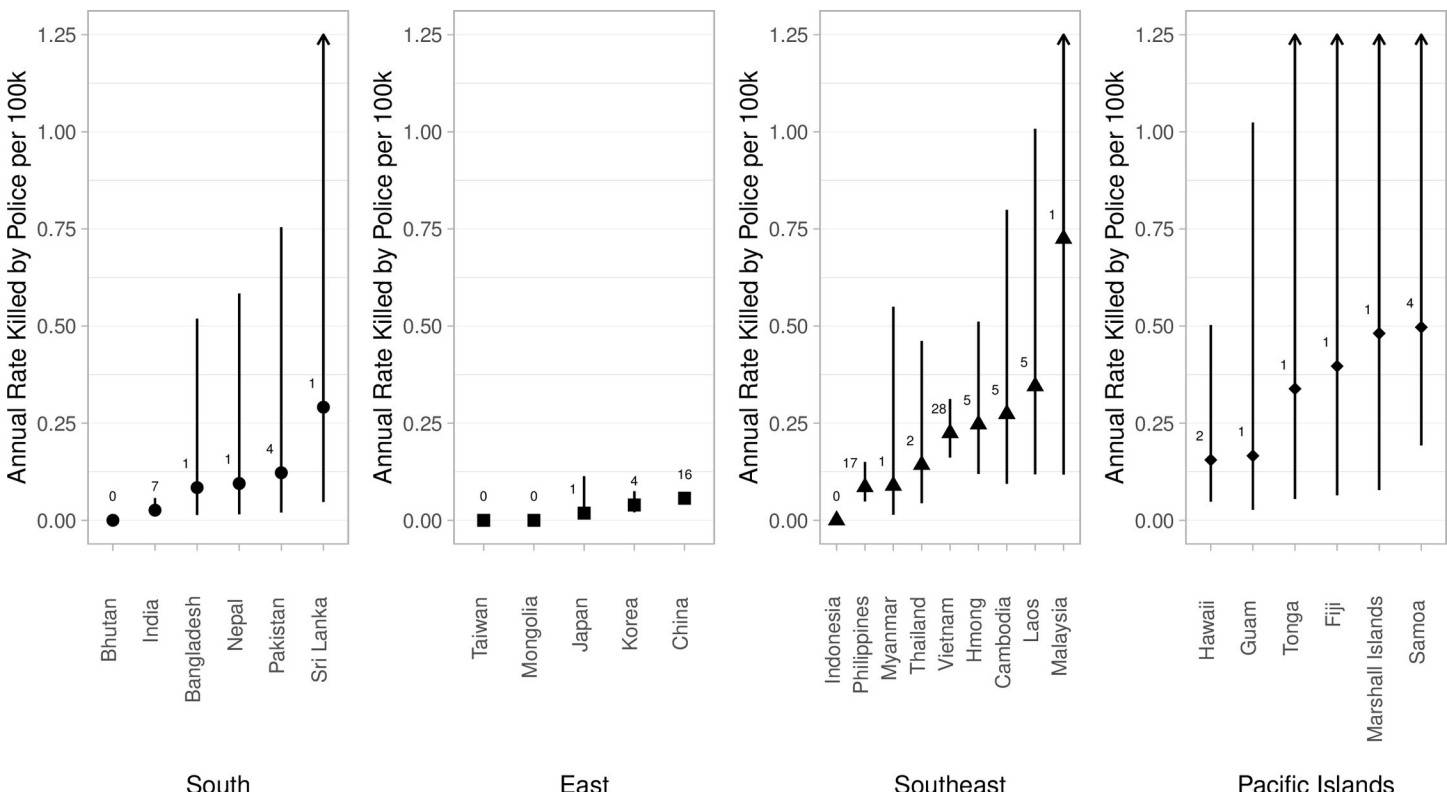

**Fig 2. Estimated annual rates of fatal police violence by AAPI national/ethnic background, 2013–2019.** Note: Vertical bars represent 95% confidence intervals. Numbers hovering above point estimates represent the number of decedents informing the estimation of each rate, calculated using a single Poisson model. Arrows indicate that confidence intervals extend beyond the Y-axis cut-off; see S4 Table in S1 Appendix. Total numerator N = 109 deaths; 55/167 API decedents were missing national/ethnic background, and an additional 3 from the Federated States of Micronesia lacked a corresponding population denominator from the US Census.

the necessity of national/ethnic data for describing inequities in risk of fatal police violence. Furthermore, our findings have broad public health importance given potential implications of fatal police violence for the health of people of the decedent's community and same racial and ethnic group, which may be even more significant in smaller ethnic groups or tight-knit ethnic communities and/or if these deaths involve immigration enforcement agents [19]. Recent studies, for example, have indicated that fatal police violence likely has spillover effects, impacting the mental, perinatal, and cardiovascular health of community members with the same racial/ethnic background as decedents [20–23]. Previous studies of racial/ethnic inequities in police violence have largely ignored national/ethnic heterogeneity within larger racial/ethnic groups [2,3,24,25], as needed data rarely exists. This omission engenders profound undertheorizing about ethnicity, immigration, and racialization in the epidemiologic study of fatal police violence. Developing that theory requires more comprehensive federal violence monitoring and Census data, including population counts for Middle Eastern/North African Americans. Future research is needed to further understand the high rates among Pacific Islanders, many of whom descend from countries subject to US colonization, and for Southeast Asian Americans whose families originated in countries heavily impacted by US war in Southeast Asia. High rates for these groups, despite large confidence intervals, suggest a trans-Pacific continuity of US state violence against Pacific Islanders and Southeast Asians that requires policy attention and political action.

The failure to disaggregate not only masks within-race inequities, but in the context of mortality caused by the criminal legal system, it reifies the "model minority myth" that AAPI

people uniformly achieve economic success and social integration and can therefore be used as a comparison group to justify anti-Black racism. Indeed, criminologists have interpreted low rates of arrests among AAPI people as evidence that the causes of high arrest rates among Black people originate within Black communities, as opposed to originating in institutional and structural racism [26]. Neglecting disaggregation may thus impede broader efforts at understanding how and why race and ethnicity matter for risk of fatal police violence and policies to eliminate racial and ethnic inequities.

Our analysis has key limitations. Most importantly, we could not identify the regional (18%) or national/ethnic (33%) background of all decedents, and because not all police violence fatalities are captured in Fatal Encounters/MPV, the presented results underestimate true rates during the study period. These undercounts are likely differential. Native Hawaiians, for example, were almost certainly undercounted due to limitations in media reports (see Methods) and imputation algorithms. Indeed, Lauderdale's surname list included only names identified as Chinese, Filipino, Japanese, Vietnamese, Korean, or Indian, while NamePrism's East/South/Southeast Asian or Pacific Islander countries were limited to Cambodia, Myanmar, Thailand, Vietnam, Indonesia, Malaysia, China/Singapore/Hong Kong/Taiwan, Philippines, South Korea, Japan, and Pakistan, Bangladesh, South Asia/Nepal/India/Sri Lanka. Many of the national/ethnic backgrounds excluded from these lists—including Laos, Hmong people, and all national/ethnic groups from the Pacific Islands—were also groups that already appear to experience among the highest rates of fatal police violence across all national/ethnic groups in this study. Thus differential undercounting likely means our core findings on the elevated risks of Pacific Islanders and Southeast Asian Americans whose families originated in areas impacted by US war in Southeast Asia are likely *under*estimated and more severe than are presented here.

Further limitations apply. The algorithms used to impute ethnicity (19% of cases) likely yielded a small number of incorrect imputations, though our checks comparing the accuracy of our imputation algorithms against our hand-codes indicate that this is likely a minor concern (see Methods). Our analysis treated all decedents as being from a single regional or national/ethnic background; media sources and identification algorithms tended to report only a single ethnicity result. This inhibited our ability to discuss the role of multiple intersecting ethnicities and likely yields mismatches between our numerators (deaths from a single ethnicity) and denominators (total single-ethnicity population for each national/ethnic group). This is a particular concern if (A) different groups have different proportions of people who identify with those groups mono-ethnically vs. multi-ethnically, or if (B) different groups were disproportionately identified as being from a single racial/ethnic group by coroners, police, or media sources when they were in fact from multiple racial groups or ethnicities. While we know that one of these problems definitively differs across groups—e.g., Pacific Islanders contain a higher proportion of people who identity as multiracial than Asian Americans [27]—we have insufficient data to assess how these two problems might *jointly* bias our estimates, given little data on the ethnicity-specific biases of reporters, coroners, or police in reporting on these deaths. Death records or media reports may also have misclassified decedents, either between AAPI groups or between races, and we relied primarily on English-language media reports and obituaries, limiting background identification. Importantly, misclassification of AAPI people as belonging to another racial group would yield underestimates for AAPI groups. Finally, the data on fatal police violence that our study draws from does not include incidents of non-fatal injury or other morbidities that occur during interactions with law enforcement. Racial and ethnic inequities in the broader population of individuals harmed during police encounters may be larger or smaller [28]. Moreover, although Fatal Encounters and MPV are the best available sources of data on fatal police violence, we cannot definitively rule out the

existence of cases that were included in our study but could possibly have occurred in the absence of police intervention. Our removal of ambiguous cases (see the last paragraph of our "Data on Fatal Police Violence" section above), however, helps ameliorate this concern.

## Conclusions and public health implications

Our database is nonetheless the most detailed and comprehensive record of the national/ethnic background of AAPI decedents of police violence in the United States, providing, for the first time, meaningful estimates of the rates at which police kill AAPIs of different national/ethnic and regional backgrounds. Little publicly available data on inequities in fatal police violence affecting national, ethnic, or regional groups subsumed within larger racial/ethnic categories—in this case, Pacific Islanders and specific Southeast Asian communities within "API"—has corresponded to a dearth of public attention, and thus of preventive action. The results presented here can guide future community and policy advocacy efforts to reduce these rates and protect communities at highest risk, as well as guide future work on racial/ethnic disaggregation in the study of fatal police violence.

## Supporting information

**S1 Appendix.**
(DOCX)

## Acknowledgments

We are grateful to D. Brian Burghart at Fatal Encounters and Samuel Sinyangwe and the other members of the Mapping Police Violence collaborative for their efforts tracking and validating the most comprehensive available databases of fatal police violence in the US.

## Author Contributions

**Conceptualization:** Gabriel L. Schwartz.

**Data curation:** Gabriel L. Schwartz, Jaquelyn L. Jahn.

**Formal analysis:** Gabriel L. Schwartz, Jaquelyn L. Jahn.

**Investigation:** Gabriel L. Schwartz, Jaquelyn L. Jahn.

**Methodology:** Gabriel L. Schwartz, Jaquelyn L. Jahn.

**Project administration:** Gabriel L. Schwartz, Jaquelyn L. Jahn.

**Software:** Gabriel L. Schwartz, Jaquelyn L. Jahn.

**Validation:** Gabriel L. Schwartz, Jaquelyn L. Jahn.

**Visualization:** Gabriel L. Schwartz, Jaquelyn L. Jahn.

**Writing – original draft:** Gabriel L. Schwartz, Jaquelyn L. Jahn.

**Writing – review & editing:** Gabriel L. Schwartz, Jaquelyn L. Jahn.

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
