## [Decision Letter · Decision Letter 0]

7 Jul 2022

PONE-D-21-36607

Disaggregating Asian American and Pacific Islander Risk of Fatal Police Violence

PLOS ONE

Dear Dr. Schwartz,

Thank you for submitting your manuscript to PLOS ONE. After careful consideration, we feel that it has merit but does not fully meet PLOS ONE’s publication criteria as it currently stands. Therefore, we invite you to submit a revised version of the manuscript that addresses the points raised during the review process.

Together, the four reviewers are expert in the substantive and methodological issues under investigation in your manuscript.

Reviewer 1 recommended major revisions and detailed a few ways you could improve some of the methodological aspects of the manuscript.

Reviewer 2 recommended minor revisions and also highlighted one or two methodological issues that you should respond to.

Reviewer 3 recommended rejection. The first comment refers to language and framing. As with something raised by Reviewer 4, this critique touches on issues raised by Nix (2020) and others. In my judgment, you should have the opportunity to respond to questions of language and framing in the revised version of the manuscript and your response to referees document.

The second issue raised by reviewer 3 regards the numerator from Fatal Encounters and Mapping Police Violence, specifically the types of deaths included, low numbers, and the proportion of individuals whose ethnic and regional background are correctly identified. The third issue regards the use of ACS population data.

As with the comments made by reviewers 1 and 2, I think you should have the opportunity to respond to these methodological matters.

Reviewer 4 also recommended rejection. Again, however, the issues raised are, in my judgement, ones that you could plausibly address in the revised version of the manuscript. You should acknowledge the complications of just focusing on fatalities (rather than also included police shootings that do not result in the death of the civilian) and explain why you include “unknown” cause of death (and consider the implications of doing so). Additionally, there are a few other methodological issues, some of which overlap with those raised by reviewer 3.

On the bases of these reviewers and my own reading of the paper, I am asking for major revisions.

We look forward to receiving your revised manuscript.

Kind regards,

Jonathan Jackson, Ph.D

Academic Editor

PLOS ONE

Reviewers' comments:

Reviewer's Responses to Questions

**Comments to the Author**

1. Is the manuscript technically sound, and do the data support the conclusions?

Reviewer #1: Partly

Reviewer #2: Yes

Reviewer #3: No

Reviewer #4: Partly

2. Has the statistical analysis been performed appropriately and rigorously? 

Reviewer #1: I Don't Know

Reviewer #2: Yes

Reviewer #3: No

Reviewer #4: I Don't Know

3. Have the authors made all data underlying the findings in their manuscript fully available?

Reviewer #1: Yes

Reviewer #2: Yes

Reviewer #3: Yes

Reviewer #4: Yes

4. Is the manuscript presented in an intelligible fashion and written in standard English?

Reviewer #1: Yes

Reviewer #2: Yes

Reviewer #3: Yes

Reviewer #4: Yes

5. Review Comments to the Author

Reviewer #1: This paper addresses a broadly important topic - the need to disaggregate groups when investigating rates of risk. Specifically, this paper focuses on the need to disaggregate people who are often lumped into the group Asian American Pacific Islander, who in reality should be recognized as distinct groups, from distinct ethnic, cultural, and geographic backgrounds. It does so by looking at rates of death at the hands of the police among people who have been grouped together in this way. I believe this paper needs major revisions before it can be published, however, because of the importance of the topic, and because the revisions, while major, I hope should not be overwhelming, I do hope that it will eventually be published.

My concerns all primarily revolve around data and methods. I wholeheartedly agree that disaggregation is important and should generally be given much greater consideration by other scholars. However, my concerns about the data and methods will require major revisions to address and are fundamental to the integrity of the argument.

First, I found the description of the coding of decedents by ethnicity and region to be insufficient in detail. In order to feel confident that the basis of the study is sound, the reader must be confident in the coding and it is not clear how confident the reader should be in this regard. How was imputation done? Was it merely taking the ethnic values from decedents who appear in both datasets but are only ethnically identified in one to fill in the values of the other or was there a more elaborate imputation scheme? Additionally, in the description of the checks on imputations, the authors mention that they used NamePrism's ethnicity predictions for those with a probability of at least 50% -- this seems like a very low cutoff. Why should the reader be confident in predictions that are at least better than chance? Finally, the issue of under identification for decedents of Hawaiian background seems like a problem that the authors don't satisfactorily address. The authors do mention that the counts in general are likely undercounts and so the rates are likely underestimates of the true rates. This would not be a problem if the undercounts were evenly distributed across groups so the error in the rates was evenly distributed - but here it is clear that the undercounts are not evenly distributed and therefore the rates for some groups are likely more underestimated than other groups in a systematic way, which would render comparison among groups problematic.

Second, I have concerns about the methodology. Poisson models are, of course, generally the standard for count data, with the population denominator used as the population at risk, or the exposure or offset variable in the Stata poisson command. The authors don't make it clear that this is what they use the denominator for, so that should be clarified. As written, it sounds as if they use the Poisson with a rate dependent variable, which I have to assume is not what they mean, therefore additional details in the description of the methods are needed. Additionally, the authors do not mention any control variables used in their model. Certainly, when showing a difference in rates, it is important to demonstrate that those differences cannot be accounted for by other factors. In this case, given the topic of death by police, I would want to see the model control for characteristics of the decedent, the circumstances of the encounter, and also for characteristics about the location where the death occurred. The differences in rates will be much more informative when presented in the adequate context accounting for all these factors.

I believe that these revisions, while major, are achievable, and that the topic merits these revisions and subsequent publication of the paper if they are adequately accomplished.

Reviewer #2: The authors examined differences in mortality rates due to police violence among Asian and Pacific Islander regional groups (South Asian, East Asian, Southeast Asian, and Pacific Islanders) and by ethnicity/country of origin. To do so, they painstakingly reviewed deaths catalogued by Fatal Encounters and Mapping Police Violence to determine decedents’ ethnicity/country of origin and associated regional group, and then compared mortality rates directly calculated from these data. The results of this analysis are important in their own right, and additionally this paper clearly demonstrates an urgent need for better data collection and more detailed disaggregation of analyses by race and ethnicity, particularly among Asian and Pacific Islander populations.

I have several minor comments:

1. Line 75: What does it mean to have corroborated MPV’s coding?

2. Lines 103-105: Were there any important trends when comparing the tabulated regions or countries of origin between the hand-coded data and the NamePrism or Lauderdale algorithms? ie, if you used these algorithms in place of the hand-coded data, would the total number of deaths assigned to any particular group been substantially higher or lower? I think this information is possibly more important than the individual-level concordance, given that it’s the total death counts within each group that inform the reported mortality rates.

3. Line 118: Were separate models fitted by race, by regional group, and by national/ethnic background?

4. Line 119 (and 23): What does it mean to fit models to “group-year files”?

5. Lines 178-183/Figure 2: I think it’s totally reasonable to highlight that the estimates for groups displaced by US wars in Southeast Asia have higher rates than other groups in the region, however, these estimates are in most cases incredibly uncertain due to small numbers, and that should also be highlighted in the text here.

6. Lines 230-234: I think more information about how individuals with multiple racial identities and/or multiple ethnic origins are categorized in each of the datasets is warranted up front in the methods section. Additionally, I think this point deserves more attention in the discussion section—what are the potential implications for the analysis of this categorization? What might the impact on the findings be of the mismatch between numerator and denominator?

Reviewer #3: Thank you for the opportunity to review the manuscript “Disaggregating Asian American and Pacific Islander Risk of Fatal Police Violence.” The manuscript is generally well-written and highlights the need to better account for subgroups within the AAPI population. With that said, I have some serious concerns about what conclusions can be drawn from the study in its current form. I humbly provide several suggestions below and hope the author(s) find them useful.

1. The authors point to high U.S. fatal police violence rates as a public health crisis. Indeed, reducing lethal encounters between police officers and citizens should be prioritized. However, how the authors frame this issue in their manuscript could use some adjustment. Over the past couple of years, terms like “epidemic” and “public health crisis” have been used to describe lethal encounters between police and citizens. However, the best estimates we have (which are admittedly far from perfect) are that civilian deaths occur in 0.0002% of all police-citizen encounters. Even if we wanted to compare officer-involved deaths to overall homicide rates solely, there are approximately 25,000 homicides annually in the U.S. To the best of our knowledge (again, readily recognizing the imperfect data), there are about 1,000 fatal officer-involved shootings annually. The vast majority of those shootings are ruled justified in response to an individual attempting to kill someone else or an officer. Even if we take a very aggressive stance and say half of those 1,000 fatal shootings are actually unjustified, that is still only 2% of all homicides within the U.S. Again, reducing lethal encounters between police officers and citizens should be a priority. However, poorly framed narratives of epidemics and public health crises are misleading and not helpful for identifying plausible policy interventions that may help.

2. How the authors construct their rates is problematic and leaves me unable to have a lot of confidence in what we can generalize from the results. I first address the numerator and then discuss the denominator.

The authors obtain the numerator from Fatal Encounters and Mapping Police Violence. While these databases can be helpful, substantial amounts of data cleaning must occur to make their data useful. I commend the authors for excluding “accidents,” vehicle collisions, etc. That certainly paints a more accurate picture of the data. However, while the authors say they took a conservative approach, they still include unknown cause of death. It seems to be the conservative approach would be to exclude those as well. Moreover, both databases include deaths simply where police officers are present. The officer(s) don’t need to necessarily be the proximate cause of death to have the case included. For example, this past week, Fatal Encounters made a statement on Twitter confirming that they would include the 19 deaths at the elementary school in Texas in their database because police were present when the deaths occurred. These deaths should not be attributed to officers when we are specifically discussing the topic of police-caused deaths. Also, these databases include instances where officers are not acting in an official capacity, such as domestic violence situations. Again, for this particular discussion, not excluding such cases paints an inaccurate picture. Perhaps no cases like these were included in this analysis, but the topic was not approached. This is especially relevant when the authors use the term “police violence.” That label suggests that an officer is the sole cause of the police-involved fatality, and the citizen’s contribution to the outcome is erased. While I am willing to agree to that framing in some circumstances, what deaths are included in the numerator need some precise explanation for that label to apply broadly.

Second, the number of cases is very small. While researchers are restricted to using the data at hand, it is hard to make broad generalizations from such small numbers, especially when the authors seemingly exclude any statistical significance tests. The authors state that the fitted Poisson models are a “reasonable” fit. Still, they do not provide any statistics for someone to assess whether they agree with the authors’ perception of what is reasonable. I am one that fervently believes the .05 significance level is arbitrary and unhelpful in social science. I am very willing to lend credence to findings above that level. However, the authors do not provide that chance to the reader, which is likely due to the small sample size we are dealing with.

Finally, NamePrism shows an 84% correct identification for national/ethnic background and 92% agreement of regional backgrounds. However, Lauderdale shows a 66% agreement. The authors describe this as reasonable fidelity. While I could be convinced by the 84% and 92% statistics, I cannot agree that 66% fidelity is reasonable fidelity, which again leads me to question how the numerator is measured.

Turning to the denominator. The authors use population data from the ACS. The vast majority of scholars who regularly study police use-of-force, as well as the DOJ and NIJ, largely agree that using simple population data is inappropriate in these types of studies. For ACS population data to be an appropriate denominator for a rate, we have to assume that everyone in the population has equal opportunity and probability of interacting with police officers. We know that is simply not true. Many minority populations have a higher probability of police interactions due to societal, system, and offending variation (just like males have a higher probability of police encounters than females). Not accounting for those differences and only using population data provides an inaccurate rate from which we cannot learn much.

Reviewer #4: The current study examined deaths at the hands of US law enforcement with a focus on Asian Americans and Pacific Islanders (AAPI) using data from Fatal Encounters (FE) and Mapping Policing Violence (MPV) from 2013-2019. I think there is some value in the hook of the paper/RQs: namely, such a large label likely masks variation across different ethnic and region groups. However, the authors continue to frame their research in the same way as other publications in this journal - despite the flaws in doing so.

1) One of the biggest issues I have is the blanket term "police violence." Nix (2020) provided a well-written response to the authors' paper in this journal. Real and valid critiques were provided there. Yet, there is no inclusion of an extremely relevant response the last time the authors published in the journal. This term is, frankly, erroneous and politically charged.

Nix, J. (2020). On the challenges associated with the study of police use of deadly force in the United States: A response to Schwartz & Jahn. PLoS one, 15(7), e0236158.

2) It is not a good idea to continue exclusively using fatalities. Nix & Shjarback's (2021) paper PLoS ONE in November 2021 showcases how much data is lost by exclusively focusing on those who die at the hands of police. Those citizens in FE and MPV (as well as The Washington Post) are likely a non-random subset of all victims of police deadly force.

Nix, J., & Shjarback, J. A. (2021). Factors associated with police shooting mortality: A focus on race and a plea for more comprehensive data. PLoS one, 16(11), e0259024.

3) What is the justification for including cases where the cause of death (COD) was "unknown" when the authors seem to contradict themselves in the same sentence? (pg. 4, lines 66-69)

4) The authors state on pg 4 "we then identified additional Asian/Pacific Islander decedents from Mapping Police Violence (MPV), a similar and overlapping database, that were not included in the Fatal Encounters list (or were listed as another race in FE) and included them if we could corroborate MPV’s coding." What was the conclusion? Similar #/% of cases included? How many additional cases from MPV?

5) There seems like a lack of certainty, especially for measuring the national/ethnic background. This may contribute to measurement error. This is even more of an issue when you consider the small base rates since the number of AAPI deaths in general are small/rare.

6) What percentage of the total relevant FE deaths do AAPI constitute? From my calculation/research, it is a small %. Is it worthy of this much investigation?

6. PLOS authors have the option to publish the peer review history of their article (what does this mean?). If published, this will include your full peer review and any attached files.

Reviewer #1: No

Reviewer #2: No

Reviewer #3: No

Reviewer #4: No

---

## [Author Response · Author response to Decision Letter 0]

9 Aug 2022

(As our response appears to be longer than what is allowable for it all be viewed in this field in the PDF, please see our attached Response Memo for full responses to Reviewer 4.)

REVIEWER #1

This paper addresses a broadly important topic - the need to disaggregate groups when investigating rates of risk. Specifically, this paper focuses on the need to disaggregate people who are often lumped into the group Asian American Pacific Islander, who in reality should be recognized as distinct groups, from distinct ethnic, cultural, and geographic backgrounds. It does so by looking at rates of death at the hands of the police among people who have been grouped together in this way. I believe this paper needs major revisions before it can be published, however, because of the importance of the topic, and because the revisions, while major, I hope should not be overwhelming, I do hope that it will eventually be published.

My concerns all primarily revolve around data and methods. I wholeheartedly agree that disaggregation is important and should generally be given much greater consideration by other scholars. However, my concerns about the data and methods will require major revisions to address and are fundamental to the integrity of the argument.

1. First, I found the description of the coding of decedents by ethnicity and region to be insufficient in detail. In order to feel confident that the basis of the study is sound, the reader must be confident in the coding and it is not clear how confident the reader should be in this regard.

a. How was imputation done? Was it merely taking the ethnic values from decedents who appear in both datasets but are only ethnically identified in one to fill in the values of the other or was there a more elaborate imputation scheme?

RESPONSE: Thank you to the reviewer for pushing us to be clearer here. We’ve now added substantial detail about how we combined information from the FE and MPV databases (revised text in green):

“Racial classification in FE uses six categories: American Indian/Alaska Native, Asian/Pacific Islander, Black, White, Hispanic, and Middle Eastern. To ensure we were not missing AAPI cases or misidentifying the race/ethnicity of AAPI decedents, we cross-referenced with Mapping Police Violence (MPV), a similar and overlapping database. No cases were identified by MPV but not by FE. In cases where MPV identified a case as AAPI but FE identified them as some other race/ethnicity (or was missing racial/ethnic data), we performed additional “hand-coding” searches to see if we could find publicly available evidence corroborating MPV’s coding (i.e., evidence that a person was in fact AAPI). Of these 13 cases, we were able to corroborate MPV’s coding in 3 cases (which were subsequently added to our list of API decedents), conclusively rejected MPV’s coding in 4 cases (not added), and were unable to corroborate nor reject MPV’s coding in 6 cases. In the latter 6 cases, we defaulted to FE’s coding. We also researched those classified as “Middle Eastern” by FE to identify misclassified South Asians; otherwise, “Middle Easterners” were coded as White in our analysis, as they are often listed as such by the Census (12) and our analysis required Census denominators. In total, this yielded 167 AAPI decedents (see below).”

b. Additionally, in the description of the checks on imputations, the authors mention that they used NamePrism's ethnicity predictions for those with a probability of at least 50% -- this seems like a very low cutoff. Why should the reader be confident in predictions that are at least better than chance?

RESPONSE: Our apologies; we now see that the way we initially wrote this section was confusing. We have amended the text to make this clearer.

Briefly, NamePrism assigns each possible ethnicity that could plausibly match a given name a score, with the scores totaling 100. We use the ethnicity with the highest score, using only those with a score over 50. As we write, when we checked these predictions against our hand-codes when using a threshold of 50, NamePrism correctly identified 84% of hand-coded national backgrounds and 92% of regional backgrounds. That is, NamePrism’s predictions with this threshold are robust, and substantially better than chance. 

Further, since only 5 participants’ national backgrounds and 6 participants’ regional backgrounds were identified via NamePrism, we would expect <1 misclassification total, using those 84% and 92% accuracy findings and multiplying by 5 and 6, respectively. Thus, empirically, there is a very low chance of NamePrism biasing our rate estimates.

New text is highlighted in green:

“To assess the accuracy of our imputations, we compared backgrounds as predicted by each algorithm (Lauderdale, NamePrism) to backgrounds as determined via hand-codes. For NamePrism, the algorithm provided a probability score for each predicted ethnicity classification (e.g., the algorithm could predict that, given its list of surnames, a given individual had a 70% chance of being Chinese, a 20% chance of being Vietnamese, and a 10% chance of being Lao). We only used NamePrism’s ethnicity predictions for those with a single ethnicity that scored at least 50. Using this cutoff, NamePrism correctly identified 84% of hand-coded national/ethnic backgrounds and 92% of hand-coded regional backgrounds.”

c. Finally, the issue of under identification for decedents of Hawaiian background seems like a problem that the authors don't satisfactorily address. The authors do mention that the counts in general are likely undercounts and so the rates are likely underestimates of the true rates. This would not be a problem if the undercounts were evenly distributed across groups so the error in the rates was evenly distributed - but here it is clear that the undercounts are not evenly distributed and therefore the rates for some groups are likely more underestimated than other groups in a systematic way, which would render comparison among groups problematic.

RESPONSE: We thank the reviewer for raising this issue. This is part of the reason we specifically called out the almost certain differential undercount of Native Hawaiians. But we agree that this point needs to be broadened. In particular, we are likely to have differentially undercounted people of national/ethnic backgrounds who are not included in the Lauderdale or NamePrism algorithms, including every Pacific Islander country. (Lauderdale only included 6 countries, which we specified explicitly, but it is also worth noting that NamePrism’s East/South/Southeast Asian or Pacific Islander countries are limited to Cambodia, Myanmar, Thailand, Vietnam, Indonesia, Malaysia, China/Singapore/Hong Kong/Taiwan, Philippines, South Korea, Japan, and Pakistan, Bangladesh, South Asia/Nepal/India/Sri Lanka).

It is worth noting, though, that the countries not included in these algorithms and the ethnicities least likely to be identified in journalistic reporting—such as Laos, Hmong people, all countries in the Pacific Islands, and Native Hawaiians—already appear to have higher rates than other groups. That is, key comparisons between groups are if anything underestimated, which would strengthen as opposed to weaken the key findings of our paper.

We have revised the text as follows:

“Our analysis has key limitations. Most importantly, we could not identify the regional (18%) or national/ethnic (33%) background of all decedents, and because not all police violence fatalities are captured in Fatal Encounters/MPV, the presented results underestimate true rates during the study period. These undercounts are likely differential. Native Hawaiians, for example, were almost certainly undercounted due to limitations in media reports (see Methods) and imputation algorithms. Indeed, Lauderdale’s surname list included only names identified as Chinese, Filipino, Japanese, Vietnamese, Korean, or Indian, while NamePrism’s East/South/Southeast Asian or Pacific Islander countries were limited to Cambodia, Myanmar, Thailand, Vietnam, Indonesia, Malaysia, China/Singapore/Hong Kong/Taiwan, Philippines, South Korea, Japan, and Pakistan, Bangladesh, South Asia/Nepal/India/Sri Lanka. Many of the national/ethnic backgrounds excluded from these lists—including not only Native Hawaiians but also Laos, Hmong people, and all countries in the Pacific Islands—were also groups that already appear to experience among the highest rates of fatal police violence across all national/ethnic groups in this study. Thus differential undercounting likely means our core findings on the elevated risks of Pacific Islanders and Southeast Asian Americans whose families originated in areas impacted by US war in Southeast Asia are likely underestimated and more severe than are presented here.”

2. Second, I have concerns about the methodology. Poisson models are, of course, generally the standard for count data, with the population denominator used as the population at risk, or the exposure or offset variable in the Stata poisson command. 

a. The authors don't make it clear that this is what they use the denominator for, so that should be clarified.

RESPONSE: This is an important point to clarify. Our Poisson models use our Census denominators as population offsets, representing the population at risk. We mentioned this briefly in our original submission but have now added more emphasis to ensure this is clear. (New text in green:)

“To calculate average annual rates of death by police violence for people of different racial, and AAPI national/ethnic and regional, backgrounds, we fit Poisson models predicting total deaths in a given group in a given year, with fixed effects for each group and corresponding population offsets (representing the total population of each group, which effectively serve as our denominators for calculating fatality rates). These were fit on “group-year” files, i.e., files in which each national/ethnic group was represented by a series of rows, each representing a different year (a “long” longitudinal file), or in which each regional background was represented by a series of such rows.”

b. As written, it sounds as if they use the Poisson with a rate dependent variable, which I have to assume is not what they mean, therefore additional details in the description of the methods are needed.

RESPONSE: Please see our responses to part (a) above and Comment 4 from Reviewer 2 below. We fit Poisson models where the outcome was total deaths for a given group in a given year, with an offset for the total population of that group in that year. Because we include an offset, our models estimate (log) rate ratios. Since we include fixed effects for each group, we essentially get a log rate ratio comparing each group to the reference and can thus can easily calculate a rate for each group.

c. Additionally, the authors do not mention any control variables used in their model. Certainly, when showing a difference in rates, it is important to demonstrate that those differences cannot be accounted for by other factors. In this case, given the topic of death by police, I would want to see the model control for characteristics of the decedent, the circumstances of the encounter, and also for characteristics about the location where the death occurred. The differences in rates will be much more informative when presented in the adequate context accounting for all these factors.

RESPONSE: We appreciate this comment. This is a descriptive paper, not an explanatory one. No individual or event-specific characteristics of the encounter could cause one to be a race or ethnicity, and so confounding is not possible. We hope merely to show variation so that future research using larger samples that more systematically track ethnicity information—or qualitative work—can better understand what is driving these disparities.

In other words, if different national/ethnic groups systematically differ in the circumstances of their encounters with the police, that would help explain the patterns we see here. But that data does not exist. Here, we provide motivation for better data collection so that more explanatory research can be undertaken.

I believe that these revisions, while major, are achievable, and that the topic merits these revisions and subsequent publication of the paper if they are adequately accomplished.

REVIEWER #2

The authors examined differences in mortality rates due to police violence among Asian and Pacific Islander regional groups (South Asian, East Asian, Southeast Asian, and Pacific Islanders) and by ethnicity/country of origin. To do so, they painstakingly reviewed deaths catalogued by Fatal Encounters and Mapping Police Violence to determine decedents’ ethnicity/country of origin and associated regional group, and then compared mortality rates directly calculated from these data. The results of this analysis are important in their own right, and additionally this paper clearly demonstrates an urgent need for better data collection and more detailed disaggregation of analyses by race and ethnicity, particularly among Asian and Pacific Islander populations.

I have several minor comments:

1. Line 75: What does it mean to have corroborated MPV’s coding?

RESPONSE: We apologize for not including more clarity. We have added text to better explain this process (new/revised text in green):

“Racial classification in FE uses six categories: American Indian/Alaska Native, Asian/Pacific Islander, Black, White, Hispanic, and Middle Eastern. To ensure we were not missing AAPI cases or misidentifying the race/ethnicity of AAPI decedents, we cross-referenced with Mapping Police Violence (MPV), a similar and overlapping database. No decedents were identified by MPV but not by FE. In cases where MPV identified a decedent as AAPI but FE identified them as some other race/ethnicity (or was missing race/ethnicity data), we performed additional “hand-coding” searches to see if we could find publicly available evidence corroborating MPV’s coding (i.e., evidence that a person was in fact AAPI). Of these 13 cases, we were able to corroborate MPV’s coding in 3 cases (which were subsequently added to our list of API decedents), conclusively rejected MPV’s coding in 4 cases (not added to our list), and were unable to corroborate nor reject MPV’s coding in 6 cases. In the latter 6 cases, we defaulted to FE’s coding.”

2. Lines 103-105: Were there any important trends when comparing the tabulated regions or countries of origin between the hand-coded data and the NamePrism or Lauderdale algorithms? ie, if you used these algorithms in place of the hand-coded data, would the total number of deaths assigned to any particular group been substantially higher or lower? I think this information is possibly more important than the individual-level concordance, given that it’s the total death counts within each group that inform the reported mortality rates.

RESPONSE: We appreciate this reviewer’s important point. We made the table below (now added to the paper’s Appendix) to assess whether certain groups were more likely to be coded using a certain algorithm. We find that across all groups, the majority of deaths were hand coded, with the exception of Chinese people, who were equally likely to be hand coded and coded by the Lauderdale method (7 deaths or 43% each). The paper now points to this table in the following text: “A breakdown of classification method across ethnic groups, showing how many decedents in each group were identified using each imputation method, is available in the Appendix.”

National/Ethnic Background Hand Coded Lauderdale Name Prism Missing

Bangladesh 1 (100.00%) 0 (0.00%) 0 (0.00%) 0 (0.00%)

Burma 1 (100.00%) 0 (0.00%) 0 (0.00%) 0 (0.00%)

Cambodia 5 (100.00%) 0 (0.00%) 0 (0.00%) 0 (0.00%)

China 7 (43.75%) 7 (43.75%) 2 (12.50%) 0 (0.00%)

Fiji 1 (100.00%) 0 (0.00%) 0 (0.00%) 0 (0.00%)

Guam 1 (100.00%) 0 (0.00%) 0 (0.00%) 0 (0.00%)

Hawaii 2 (100.00%) 0 (0.00%) 0 (0.00%) 0 (0.00%)

Hmong 5 (100.00%) 0 (0.00%) 0 (0.00%) 0 (0.00%)

India 4 (57.14%) 3 (42.86%) 0 (0.00%) 0 (0.00%)

Japan 0 (0.00%) 1 (100.00%) 0 (0.00%) 0 (0.00%)

Korea 3 (75.00%) 1 (25.00%) 0 (0.00%) 0 (0.00%)

Laos 5 (100.00%) 0 (0.00%) 0 (0.00%) 0 (0.00%)

Malaysia 1 (100.00%) 0 (0.00%) 0 (0.00%) 0 (0.00%)

Marshall Islands 1 (100.00%) 0 (0.00%) 0 (0.00%) 0 (0.00%)

Micronesia 3 (100.00%) 0 (0.00%) 0 (0.00%) 0 (0.00%)

Nepal 1 (100.00%) 0 (0.00%) 0 (0.00%) 0 (0.00%)

Pakistan 4 (100.00%) 0 (0.00%) 0 (0.00%) 0 (0.00%)

Philippines 12 (70.59%) 4 (23.53%) 1 (5.88%) 0 (0.00%)

Samoa 4 (100.00%) 0 (0.00%) 0 (0.00%) 0 (0.00%)

Sri Lanka 1 (100.00%) 0 (0.00%) 0 (0.00%) 0 (0.00%)

Thailand 2 (100.00%) 0 (0.00%) 0 (0.00%) 0 (0.00%)

Tonga 1 (100.00%) 0 (0.00%) 0 (0.00%) 0 (0.00%)

Vietnam 16 (57.14%) 10 (35.71%) 2 (7.14%) 0 (0.00%)

Missing 0 (0.00%) 0 (0.00%) 0 (0.00%) 31 (100.00%)

We have also included a more detailed paragraph in the discussion, unpacking what this means for our estimates (new/revised text in green):

“Our analysis has key limitations. Most importantly, we could not identify the regional (18%) or national/ethnic (33%) background of all decedents, and because not all police violence fatalities are captured in Fatal Encounters/MPV, the presented results underestimate true rates during the study period. These undercounts are likely differential. Native Hawaiians, for example, were almost certainly undercounted due to limitations in media reports (see Methods) and imputation algorithms. Indeed, Lauderdale’s surname list included only names identified as Chinese, Filipino, Japanese, Vietnamese, Korean, or Indian, while NamePrism’s East/South/Southeast Asian or Pacific Islander countries were limited to Cambodia, Myanmar, Thailand, Vietnam, Indonesia, Malaysia, China/Singapore/Hong Kong/Taiwan, Philippines, South Korea, Japan, and Pakistan, Bangladesh, South Asia/Nepal/India/Sri Lanka. Many of the national/ethnic backgrounds excluded from these lists—including not only Native Hawaiians but also Laos, Hmong people, and all countries in the Pacific Islands—were also groups that already appear to experience among the highest rates of fatal police violence across all national/ethnic groups in this study. Thus differential undercounting likely means our core findings on the elevated risks of Pacific Islanders and Southeast Asian Americans whose families originated in areas impacted by US war in Southeast Asia are likely underestimated and more severe than are presented here.”

3. Line 118: Were separate models fitted by race, by regional group, and by national/ethnic background?

RESPONSE: This is a critical point which is core to our methods; we have taken the reviewer’s feedback to heart and made this clearer. Shortly, no: we fit one model for all national/ethnic backgrounds together (as visualized in Figure 2), and a second model for all regional groups and other racial/ethnic groups to whom it was useful to make comparisons (as visualized in Figure 1). We rephrased the modeling section of our study’s Methods to more clearly state that we predicted total deaths (across all ethnic groups) in the model (new/revised text in green). We also added to our Figure notes to make sure this was also clear there.

“To calculate average annual rates of death by police violence for people of different racial, and AAPI national/ethnic and regional, backgrounds, we fit robust Poisson models predicting total deaths in a given group in a given year, with fixed effects for each group and corresponding population offsets (representing the total population of each group, which effectively serve as our denominators for calculating fatality rates). These consisted of a single model predicting all national/ethnic group rates, and a second model predicting all regional rates (as well as rates for other racial/ethnic groups, for comparison). These were fit on “group-year” files, i.e., files in which each national/ethnic group was represented by a series of rows, each representing a different year (a “long” longitudinal file), or in which each regional background was represented by a series of such rows.

4. Line 119 (and 23): What does it mean to fit models to “group-year files”?

RESPONSE: Apologies for the miscommunication. This was meant to describe our data structure, so that readers could better understand how our models were estimated. We have now clarified this in the text (see quoted text from our response to Comment 3 by this reviewer), as well in the figure notes for each figure.

5. Lines 178-183/Figure 2: I think it’s totally reasonable to highlight that the estimates for groups displaced by US wars in Southeast Asia have higher rates than other groups in the region, however, these estimates are in most cases incredibly uncertain due to small numbers, and that should also be highlighted in the text here.

RESPONSE: We agree this is important to highlight, and have done so by revising the following sentences:

“Among Southeast Asian Americans, some rates were near zero, while national/ethnic groups displaced by US war in Southeast Asia (Vietnamese, Cambodian, Lao, and Hmong) had rates between 0.22 and 0.35, although confidence intervals for these estimates were wide and sometimes overlapped with those of other national/ethnic groups.”

“High rates for these groups, despite large confidence intervals, suggest a trans-Pacific continuity of US state violence against Pacific Islanders and Southeast Asians that requires greater policy attention and political action.”

We note, though, that even with large confidence intervals, groups like Vietnamese Americans are statistically significantly distinct from groups like Indonesian or Filipino people, or any national/ethnic group in East Asia, and thus we believe are worth highlighting.

6. Lines 230-234: I think more information about how individuals with multiple racial identities and/or multiple ethnic origins are categorized in each of the datasets is warranted up front in the methods section. Additionally, I think this point deserves more attention in the discussion section—what are the potential implications for the analysis of this categorization? What might the impact on the findings be of the mismatch between numerator and denominator?

RESPONSE: We agree, and have now expanded on this in our discussion. Particularly, there is unfortunately insufficient data to clearly state how these biases might jointly affect our estimates, but it is worth stating this explicitly (new/revised text in green):

“Our analysis treated all decedents as being from a single regional or national/ethnic background; media sources and identification algorithms tended to report only a single ethnicity result. This inhibited our ability to discuss the role of multiple intersecting ethnicities and likely yields mismatches between our numerators (deaths from a single ethnicity) and denominators (total population for each national/ethnic group). This is a particular concern if (A) different groups have different proportions of people who identify with those groups mono-ethnically vs. multi-ethnically, or if (B) different groups were disproportionately identified as being from a single racial/ethnic group by coroners, police, or media sources when they were in fact from multiple racial groups or ethnicities. While we know that one of these problems definitively differs across groups—e.g., Pacific Islanders contain a higher proportion of people who identity as multiracial than Asian Americans—we have insufficient data to assess how these two problems might jointly bias our estimates, given little data on the ethnicity-specific biases of reporters, coroners, or police in reporting on these deaths.”

REVIEWER #3

Thank you for the opportunity to review the manuscript “Disaggregating Asian American and Pacific Islander Risk of Fatal Police Violence.” The manuscript is generally well-written and highlights the need to better account for subgroups within the AAPI population. With that said, I have some serious concerns about what conclusions can be drawn from the study in its current form. I humbly provide several suggestions below and hope the author(s) find them useful.

1. The authors point to high U.S. fatal police violence rates as a public health crisis. Indeed, reducing lethal encounters between police officers and citizens should be prioritized. However, how the authors frame this issue in their manuscript could use some adjustment. Over the past couple of years, terms like “epidemic” and “public health crisis” have been used to describe lethal encounters between police and citizens. However, the best estimates we have (which are admittedly far from perfect) are that civilian deaths occur in 0.0002% of all police-citizen encounters. Even if we wanted to compare officer-involved deaths to overall homicide rates solely, there are approximately 25,000 homicides annually in the U.S. To the best of our knowledge (again, readily recognizing the imperfect data), there are about 1,000 fatal officer-involved shootings annually. The vast majority of those shootings are ruled justified in response to an individual attempting to kill someone else or an officer. Even if we take a very aggressive stance and say half of those 1,000 fatal shootings are actually unjustified, that is still only 2% of all homicides within the U.S. Again, reducing lethal encounters between police officers and citizens should be a priority. However, poorly framed narratives of epidemics and public health crises are misleading and not helpful for identifying plausible policy interventions that may help.

RESPONSE: We frame fatal police violence as a “crisis” only in the paper’s abstract, and have rephrased that sentence to read, “High rates and racial inequities in U.S. fatal police violence are an urgent area of public health concern and policy attention.” We believe this language is justified given that the American Public Health Association, the nation’s largest network of public health professionals, released a policy statement that identifies fatal police violence as an important public health problem and describes policy solutions:

https://www.apha.org/policies-and-advocacy/public-health-policy-statements/policy-database/2019/01/29/law-enforcement-violence

2. How the authors construct their rates is problematic and leaves me unable to have a lot of confidence in what we can generalize from the results. I first address the numerator and then discuss the denominator.

a. The authors obtain the numerator from Fatal Encounters and Mapping Police Violence. While these databases can be helpful, substantial amounts of data cleaning must occur to make their data useful. I commend the authors for excluding “accidents,” vehicle collisions, etc. That certainly paints a more accurate picture of the data. However, while the authors say they took a conservative approach, they still include unknown cause of death. It seems to be the conservative approach would be to exclude those as well.

RESPONSE: We appreciate this reviewer’s concern. First, it is worth noting that only two cases had an unknown cause of death, which we now point out in our manuscript. One of those two was missing all background information after our hand-coding and imputation, and so was not included in any rate calculation; he was, however, killed in Hawai’i, increasing the likelihood that he was a Pacific Islander. The other was Samoan.

Including or excluding these cases in the former case would not change our rate estimates at all, and in the other case would not change them substantively: that Pacific Islanders as a whole were killed at rates similar to AIAN and Black people in the US, and that specific Pacific Islander national/ethnic groups appeared to have high rates but rates for those specific groups were extremely imprecisely estimated.

Generally, however, we disagree that including cases with an uncertain cause of death is over-reach. In general, we remove cases where there is evidence or testimony that a decedent’s death was not caused by any weapon police are armed with, or by the police in some other way. In these two cases, we lack such evidence, and therefore include these two cases. We assert that this is a reasonable choice of scientific inclusion criteria. In any case, it would not change our conclusions, and thus hope this gives the reviewer some confidence that this is a minor concern in any case.

b. Moreover, both databases include deaths simply where police officers are present. The officer(s) don’t need to necessarily be the proximate cause of death to have the case included. For example, this past week, Fatal Encounters made a statement on Twitter confirming that they would include the 19 deaths at the elementary school in Texas in their database because police were present when the deaths occurred. These deaths should not be attributed to officers when we are specifically discussing the topic of police-caused deaths. Also, these databases include instances where officers are not acting in an official capacity, such as domestic violence situations. Again, for this particular discussion, not excluding such cases paints an inaccurate picture. Perhaps no cases like these were included in this analysis, but the topic was not approached. This is especially relevant when the authors use the term “police violence.” That label suggests that an officer is the sole cause of the police-involved fatality, and the citizen’s contribution to the outcome is erased. While I am willing to agree to that framing in some circumstances, what deaths are included in the numerator need some precise explanation for that label to apply broadly.

RESPONSE: This is a serious concern and one we considered and dealt with carefully. We did not, however, discuss how we dealt with this in our initial submission, and we thank the reviewer for pushing us to explicitly address this issue in writing. As we read extensive news coverage about every single AAPI death in our database, we were able to purposefully locate and remove these cases. We’ve now added the following text to our discussion to describe that process:

“Importantly, though rare, Fatal Encounters can include deaths in police custody that were in fact killings performed by other people during a confrontation with police. (For example, take the specific case in our initial list where Person A had a gun and was holding Person B hostage, then police killed Person A; but Person B appeared to have been killed by Person A prior to the police engaging Person A. FE would include both Person A and Person B in their database.) Since, through our hand-coding, we read extensive news coverage on every AAPI death included in FE, we were able to remove any cases equivalent to Person B. We found 4 such cases, who were not included in our final analytic list.”

c. Second, the number of cases is very small. While researchers are restricted to using the data at hand, it is hard to make broad generalizations from such small numbers, especially when the authors seemingly exclude any statistical significance tests. The authors state that the fitted Poisson models are a “reasonable” fit. Still, they do not provide any statistics for someone to assess whether they agree with the authors’ perception of what is reasonable. I am one that fervently believes the .05 significance level is arbitrary and unhelpful in social science. I am very willing to lend credence to findings above that level. However, the authors do not provide that chance to the reader, which is likely due to the small sample size we are dealing with.

RESPONSE: There are two important questions here. The first is whether we include information about uncertainty, allowing statistically meaningful comparisons between rates. The second is an assessment of the quality of our Poisson models’ fit. We address each separately.

First, we do include uncertainty estimates, as reflected in our 95% confidence intervals. The size of these CIs are reflective of the small numbers of these incidents. We include these CIs both in our figures and in our appendix, detailing their exact bounds. Comparing 95% confidence intervals provides a useful heuristic: if CIs between groups did not overlap, groups are statistically significantly different. For example, Pacific Islanders appear to definitively differ in their rates (as calculated by our models) from the rates experienced by people of other AAPI regional backgrounds, and Vietnamese people clearly differ from the rates among Indonesian, Filipino, or any East Asian group. There are cases where a pair-wise comparison would show statistically significant differences between two groups even when their CIs slightly overlap; but comparing each national/ethnic group to every other national/ethnic group would require 325 pairwise comparisons and create untenable multiple testing problems. We thus simply provide estimates and CIs for each group, allowing meaningful inferences about between-group comparisons.

Second, we have added additional information about goodness of fit (new/revised text below is in green). Notably, we get nearly identical estimates from negative binomial models, which underscores that different distributional assumptions make little difference for our conclusions. We also moved this section down into our Results section, since reporting specific goodness of fit statistics seemed most suitable there:

“Goodness-of-fit statistics indicated regular Poisson models were a reasonable fit to these data (for national/ethnic background: deviance goodness-of-fit χ^2: p=0.930, Pearson goodness-of-fit χ^2: p=0.533, Wald test of improved fit over a null model: p<0.0001; for regional background, deviance goodness-of-fit χ^2: p=0.656, Pearson goodness-of-fit χ^2: p=0.747, Wald test of improved fit over a null model: p<0.0001;). Estimates calculated via negative binomial models were functionally identical (and α values assessing overdispersion were effectively 0 for both models).”

d. Finally, NamePrism shows an 84% correct identification for national/ethnic background and 92% agreement of regional backgrounds. However, Lauderdale shows a 66% agreement. The authors describe this as reasonable fidelity. While I could be convinced by the 84% and 92% statistics, I cannot agree that 66% fidelity is reasonable fidelity, which again leads me to question how the numerator is measured.

RESPONSE: In response to this and Reviewer 2’s comment above, we now include a table in the Appendix showing the percentage of cases for each ethnic group identified via hand coding, Lauderdale, and Name Prism. The table suggests that for nearly all national ethnic groups (except Chinese people), the majority of deaths were hand-coded. Our coding scheme prioritizes the Lauderdale over NamePrism because this method has been validated in other health data:

Wong EC, Palaniappan LP, Lauderdale DS. Using name lists to infer Asian racial/ethnic subgroups in the healthcare setting. Medical Care. 2010 Jun;48(6):540-546. DOI: 10.1097/mlr.0b013e3181d559e9. PMID: 20421828; PMCID: PMC3249427.

In addition, we re-coded the ethnicity classifications prioritizing NamePrism over the Lauderdale method and show the results below with differences bolded. This revised coding scheme is not likely to affect our study’s findings because 1) the groups with the highest rates are unaffected, and 2) changes are limited to groups with the largest population denominators. 

Country Region Recoded Original

Bangladesh South 1 1

Burma Southeast 1 1

Cambodia Southeast 5 5

China East 17 16

Fiji Pacific Islander 1 1

Guam Pacific Islander 1 1

Hawaii Pacific Islander 2 2

Hmong Southeast 5 5

India South 7 7

Japan East 1 1

Korea East 3 4

Laos Southeast 5 5

Malaysia Southeast 1 1

Marshall Islands Pacific Islander 1 1

Micronesia Pacific Islander 3 3

Nepal South 1 1

Pakistan South 4 4

Philippines Southeast 18 17

Samoa Pacific Islander 4 4

Sri Lanka South 1 1

Thailand Southeast 2 2

Tonga Pacific Islander 1 1

Vietnam Southeast 28 28

Missing Pacific Islander 24 24

Missing Missing 30 31

We’ve included the following text at the close of the Results to provide readers with more information about this analysis:

“To check how changes in the order of our imputation strategy may have affected results, we reversed the order in which we applied our imputation algorithms: instead of applying Lauderdale and then applying NamePrism to cases that remained un-imputed, we did the reverse. This changed our coding of 3 total cases, yielding one additional Chinese case, one additional Filipino case, and one fewer Korean case. All of these were among our groups with the lowest number of deaths and largest denominators, meaning these changes made little difference to our estimates nor did they change any of our central conclusions.”

3. Turning to the denominator. The authors use population data from the ACS. The vast majority of scholars who regularly study police use-of-force, as well as the DOJ and NIJ, largely agree that using simple population data is inappropriate in these types of studies. For ACS population data to be an appropriate denominator for a rate, we have to assume that everyone in the population has equal opportunity and probability of interacting with police officers. We know that is simply not true. Many minority populations have a higher probability of police interactions due to societal, system, and offending variation (just like males have a higher probability of police encounters than females). Not accounting for those differences and only using population data provides an inaccurate rate from which we cannot learn much.

RESPONSE: We appreciate this reviewer’s concerns about our denominator. This is an ongoing debate in the field. We come down on a different side of that debate than this reviewer, in particular because we have a different read of (1) the literature, (2) who is at risk of fatal police violence, (3) whether factors such as a higher probability of police interaction are confounders vs. mediators, and (4) what the methodological drawbacks are of selecting on things like police stops or criminality, in that doing so induces collider stratification bias and artificially reduces racial inequities by controlling for a mediating path. We address each of those below.

First, different literature in this field uses different denominators, which ultimately means they are answering different questions (i.e., what is this population’s risk of fatal police violence? vs. what is this population’s risk of fatal police violence given that they were stopped, or given that they are engaging in criminalized activities?). We are certainly not the only or first research team to use population denominators in studies of police violence rates, nor are we the only ones to advocate for their use, including Edwards, Esposito, Mummolo, and Knox:

Edwards, F., Lee, H., & Esposito, M. (2019). Risk of being killed by police use of force in the United States by age, race–ethnicity, and sex. Proceedings of the National Academy of Sciences, 116(34), 16793-16798. doi:doi:10.1073/pnas.1821204116

Esposito, M., Larimore, S., & Lee, H. (2021). Aggressive Policing, Health, And Health Equity. Health Affairs. doi:10.1377/hpb20210412.997570

Knox, D., & Mummolo, J. (2020). Making inferences about racial disparities in police violence. Proceedings of the National Academy of Sciences, 117(3), 1261-1262. doi:doi:10.1073/pnas.1919418117

Knox, D., Lowe, W., & Mummolo, J. (2020). Administrative Records Mask Racially Biased Policing. American Political Science Review, 114(3), 619-637. doi:10.1017/S0003055420000039

Second, it is not correct that people who are, for example, engaged in criminal activity or are of a particular age range are the only ones at risk of police violence. To take obvious examples, Tamir Rice was a child playing with a toy; Elijah McClain was simply perceived to be overly excitable in a way that was “suspicious;” Breonna Taylor was asleep in her house and was not even suspected of any criminal activity. Whether individual cases were “justified” or not is not our argument here, but rather that in fact the entire population, regardless of their behavior, has a non-zero risk of being the victim of fatal police violence, and thus that population denominators are appropriate.

Third, this is a descriptive paper, not an explanatory one. While different groups have different “probabilities of police interactions due to societal, system, and offending variation,” that is a hypothetical explanation for the variation we see; it is not confounding. No individual or event-specific characteristics of the encounter could cause one to be one race or ethnicity or another, and so confounding by these characteristics is not possible. We merely show variation so that future research using larger samples that more systematically track ethnicity information—or qualitative work—can better understand what is driving these disparities. For example, it is possible that employment discrimination and higher rates of poverty among Vietnamese refugees are the drivers of, say, increased rates of illegal drug use as a coping mechanism or participation in illegal means of economic survival for this group. But that would make these mediators: being a Vietnamese refugee leads to increased exposure to racism and poorer economic prospects, which leads to activities that attract police attention, which leads to higher risk of fatal police violence. But that would make the circumstances of Viet people’s interactions with police a mediator that explains the patterns we show here, not a confounder. In our analysis, we provide motivation for better data collection so that more explanatory research of the kind this reviewer is suggesting can be undertaken.

Fourth, Knox and Mummolo show that controlling for things like “criminality” or perceived suspiciousness by police actually induces bias. If there are common causes of criminality/being stopped/etc. and risk of being killed by police, then controlling for criminality or some equivalent marker actually opens backdoor paths between race/ethnicity and fatality risk. This is illustrated in the DAG below, reproduced from Knox & Mummolo; “U” represents criminality/suspicion/circumstances. The backdoor path is: D � M � U � Y when controlling for M; U in this diagram could represent many different variables.

(See attached response.)

Even in the absence of this confounding (no U), if officers are motivated by racism to stop darker-skinned Vietnamese people (for example) than lighter-skinned Japanese people on the basis of the color of their skin, then controlling for disproportionate rates of police stops (for example) would artificially reduce the magnitude of ethnic disparities in fatal police violence by assuming that differences in stop rates are “natural” or “given” when they are in fact modifiable and reflective of colorist racism. Thus controlling for “criminality” or “stops” would inappropriately bias estimates of racial disparities towards the null. Special mediation methods are required to account for this problem; these are not possible to fit with the data we have available. For more detail, we direct the reviewer to Knox, D., Lowe, W., & Mummolo, J. (2020). Administrative Records Mask Racially Biased Policing. American Political Science Review, 114(3), 619-637. doi:10.1017/S0003055420000039.

REVIEWER #4

The current study examined deaths at the hands of US law enforcement with a focus on Asian Americans and Pacific Islanders (AAPI) using data from Fatal Encounters (FE) and Mapping Policing Violence (MPV) from 2013-2019. I think there is some value in the hook of the paper/RQs: namely, such a large label likely masks variation across different ethnic and region groups. However, the authors continue to frame their research in the same way as other publications in this journal - despite the flaws in doing so.

1. One of the biggest issues I have is the blanket term "police violence." Nix (2020) provided a well-written response to the authors' paper in this journal. Real and valid critiques were provided there. Yet, there is no inclusion of an extremely relevant response the last time the authors published in the journal. This term is, frankly, erroneous and politically charged.

Nix, J. (2020). On the challenges associated with the study of police use of deadly force in the United States: A response to Schwartz & Jahn. PLoS one, 15(7), e0236158.

RESPONSE: We appreciate this reviewer’s careful reading of Nix’s article. We disagree, however, as to the use of this language, as it is in line with the public health literature on this topic and, we argue, not more political than using the term Nix prefers, e.g. “police-involved fatalities.” We have, however, made a number of revisions to respond as directly as possible to these concerns. The last paragraph of this response speaks to the question of our language being “politically charged.”

First, we believe our choices are a reasonable scientific description of these deaths, but agree we need more clarity given the different ways different fields (e.g., public health vs. criminology) use the term “police violence.” To avoid confusion about what we mean by “violence” given the diverse readership of PLOS One, our revised manuscript now provides a definition of “fatal police violence” in the first sentence of the manuscript (pasted below). We note that this definition differs from the Philip Stinson definition mentioned in Nix’s commentary in that it makes no distinction as to whether the fatality was “justified” or not because as a matter of public health monitoring it is important to describe population distributions of these deaths regardless of whether they are considered “justified” by police, courts, or the general public. In other words, whether or not violence was a “justified” or “allowed” use of force does not mean it is not an act of violence, as Nix also acknowledges in his commentary. The point, in our view, is that structural factors make it more likely that certain groups will die from this form of violence, and that policies should be put in place to prevent this violence from occurring – whether that means (A) interventions to prevent communities from coming in contact with police because those interventions have helped prevent activities that attract police attention (e.g., economic development, increasing addiction recovery or mental health care access, etc.), or whether that means (B) efforts to change how police are instructed or allowed to use force, etc. To clear up any potential confusion, we’ve added this definition in the first sentence of our manuscript:

“Recent advances in the US’s monitoring of fatal police violence, defined here as fatalities in police custody or involving the police that would not have occurred in the absence of police intervention, have enabled a more accurate accounting of these deaths, especially with respect to racial inequities (1,2).”

We further describe the classification of these deaths in the Methods section and reference studies by Edwards, Lee & Esposito, 2019 which used the same criteria and the term “police violence:” 

“We further conservatively restricted to those who were lethally shot, tasered, asphyxiated, beaten, or whose cause of death was unknown, broadly excluding causes of death that could be considered “accidents” or that would also have occurred in the absence of police intervention (e.g., falling from a height, vehicle collisions) (2,3).”

We in fact go a step further in this paper, using our extensive reading of newspaper articles about every single death of AAPI people to remove any ambiguous cases that may have not been the result of police action:

“Importantly, though rare, Fatal Encounters can include deaths in police custody that were in fact killings performed by other people during a confrontation with police. (For example, take the specific case in our initial list where Person A had a gun and was holding Person B hostage, then police killed Person A; but Person B appeared to have been killed by Person A prior to the police engaging Person A. FE would include both Person A and Person B in their database.) Since, through our hand-coding, we read extensive news coverage on every AAPI death included in FE, we were able to remove any cases equivalent to Person B. We found 4 such cases, who were not included in our final analytic list.”

As noted above, our differences in terminology partially reflect differences between our field (public health) and Nix’s (criminology). Our use of the term “police violence” is meant to be a straightforward description aligning with its use in public health academic literature, including in major medical and public health journals such as JAMA, Preventive Medicine, American Journal of Public Health, Obstetrics and Gynecology, Journal of Urban Health, and the Proceedings of the National Academy of Sciences: 

1. Fedina L, Backes BL, Jun HJ, Shah R, Nam B, Link BG, et al. Police violence among women in four U.S. cities. Prev Med (Baltim). 2018;106: 150–156. doi:10.1016/j.ypmed.2017.10.037

2. Weed JC. Connecting Police Violence with Reproductive Health. Obstetrics and Gynecology. Lippincott Williams and Wilkins; 2017. p. 1140. doi:10.1097/AOG.0000000000002086

3. DeVylder JE, Jun HJ, Fedina L, Coleman D, Anglin D, Cogburn C, et al. Association of Exposure to Police Violence With Prevalence of Mental Health Symptoms Among Urban Residents in the United States. JAMA Netw open. 2018;1: e184945. doi:10.1001/jamanetworkopen.2018.4945

4. Calvert CM, Brady SS, Jones-Webb R. Perceptions of Violent Encounters between Police and Young Black Men across Stakeholder Groups. J Urban Heal. 2020 [cited 7 Feb 2020]. doi:10.1007/s11524-019-00417-6

5. Fehrenbacher AE, Park JN, Footer KHA, Silberzahn BE, Allen ST, Sherman SG. Exposure to Police and Client Violence Among Incarcerated Female Sex Workers in Baltimore City , Maryland. Am J Public Health. 2020;110: 152–159. doi:10.2105/AJPH.2019.305451

6. Knox D, Mummolo J. Making inferences about racial disparities in police violence. Proceedings of the National Academy of Sciences of the United States of America. NLM (Medline); 2020. pp. 1261–1262. doi:10.1073/pnas.1919418117

Still, we cannot rule out that some portion of these deaths would have occurred even in the absence of police intervention. We’ve thus 1) added the language below to the limitations section of our paper’s Discussion, and 2) removed language that implies certainty that these deaths would not have occurred in the absence of police intervention, e.g. “killed by the police”:

“Moreover, although Fatal Encounters and MPV are the best available sources of data on fatal police violence we cannot definitively rule out the existence of cases that were included in our study but could possibly have occurred in the absence of police intervention. Our removal of ambiguous cases (see the last paragraph of our “Data on Fatal Police Violence” section above), however, helps ameliorate this concern.”

Finally, it is important to speak directly to the question of whether calling these deaths instances of “fatal police violence” is “politically charged.” Many journalists have pointed out that terms like “police-involved fatality” or “police-involved shooting” are also politically charged because their use of the passive voice obscures any potential culpability on the part of police. As Leary writes, "The phrase [‘officer-involved shooting’] has real consequences for how the reading public understands police shootings. Its passive voice obscures agency and avoids even the question of culpability; there is no action, only ‘involvement’… when it circulates in the press, a police shooting, by definition a matter of power, becomes a question merely of procedure.” Wesley Lowery, a Pulitzer Prize-winning correspondent for the Washington Post, similarly referred to a move away from euphemisms such as “officer-involved shooting” as an important “reckoning over objectivity”: “[a false sense of neutrality] insists we use clunky euphemisms like ‘officer-involved shooting,’” but “moral clarity, and a faithful adherence to grammar and syntax, would demand we use words that most precisely mean the thing we’re trying to communicate: ‘the police shot someone.’” While this reviewer may think “fatal police violence” is the wording with more political charge, we would argue “police-involved fatality” is just as, if not more, charged: all proposed language choices in describing these deaths are political. We thus chose the option that best suited the public health literature and which followed the objective writing style guide recommendations of the Associated Press (see: https://twitter.com/apstylebook/status/1298283084631150592). 

For more on the political nature of “police-involved fatality,” please see:

Lowery, Wesley. (June 23, 2020). “A reckoning over objectivity, led by Black journalists.” The New York Times. Available at: https://www.nytimes.com/2020/06/23/opinion/objectivity-black-journalists-coronavirus.html. Accessed August 3, 2022.

Soderberg, Brandon; Friedman, Andy. (January 14, 2022). Major media outlets can’t stop describing police violence as “officer-involved” incidents. Huffington Post. Available at: https://www.huffpost.com/entry/police-violence-officer-involved-analysis-lapd_n_61df310fe4b0a26702885448. Accessed August 3, 2022.

Leary, John Patrick. (September 1, 2016). “Officer-involved” obfuscation. Jacobin. Available at: https://jacobin.com/2016/09/eula-love-officer-involved-shooting-black-lives-matter. Accessed August 3, 2022.

Assar, Vijith. (September 3, 2015). An interactive guide to ambiguous grammar. McSweeney’s. Available at: https://www.mcsweeneys.net/articles/an-interactive-guide-to-ambiguous-grammar. Accessed August 3, 2022.

2. It is not a good idea to continue exclusively using fatalities. Nix & Shjarback's (2021) paper PLoS ONE in November 2021 showcases how much data is lost by exclusively focusing on those who die at the hands of police. Those citizens in FE and MPV (as well as The Washington Post) are likely a non-random subset of all victims of police deadly force.

Nix, J., & Shjarback, J. A. (2021). Factors associated with police shooting mortality: A focus on race and a plea for more comprehensive data. PLoS one, 16(11), e0259024.

RESPONSE: We agree that a limitation of the Fatal Encounters data is that it does not include non-fatal injuries that occur in police encounters, and now state this in our paper’s discussion and cite the above referenced paper:

“Finally, the data on fatal police violence that our study draws from does not include incidents of non-fatal injury or other morbidities that occur during interactions with law enforcement. Racial and ethnic inequities in the broader population of individuals harmed during police encounters may be larger or smaller (28).” 

3. What is the justification for including cases where the cause of death (COD) was "unknown" when the authors seem to contradict themselves in the same sentence? (pg. 4, lines 66-69)

RESPONSE: Please refer to our response to Reviewer 3, comment 2a.

4. The authors state on pg 4 "we then identified additional Asian/Pacific Islander decedents from Mapping Police Violence (MPV), a similar and overlapping database, that were not included in the Fatal Encounters list (or were listed as another race in FE) and included them if we could corroborate MPV’s coding." What was the conclusion? Similar #/% of cases included? How many additional cases from MPV?

RESPONSE: Since this is a key point, we have revised the text to include additional information: 

“To ensure we were not missing AAPI cases or misidentifying the race/ethnicity of AAPI decedents, we cross-referenced with Mapping Police Violence (MPV), a similar and overlapping database. No cases were identified by MPV but not by FE. In cases where MPV identified a case as AAPI but FE identified them as some other race/ethnicity (or was missing racial/ethnic data), we performed additional “hand-coding” searches to see if we could find publicly available evidence corroborating MPV’s coding (i.e., evidence that a person was in fact AAPI). Of these 13 cases, we were able to corroborate MPV’s coding in 3 cases (which were subsequently added to our list of API decedents), conclusively rejected MPV’s coding in 4 cases (not added), and were unable to corroborate nor reject MPV’s coding in 6 cases. In the latter 6 cases, we defaulted to FE’s coding.”

5. There seems like a lack of certainty, especially for measuring the national/ethnic background. This may contribute to measurement error. This is even more of an issue when you consider the small base rates since the number of AAPI deaths in general are small/rare.

RESPONSE: We appreciate this concern; our main approach has been to provide key details on the degree of uncertainty for each portion of our data.

Our denominators come from the US Census, the definitive source of population counts for different groups in the United States. Thus there is little uncertainty about our denominator. For more detail about our denominator choice, please see our reply to Reviewer 3, Comment 3.

In terms of our identified cases, uncertainty in identification is, we argue, fairly low. Our hand-coded cases are based only on clear indications in media or obituaries of someone’s ethnic background. Our imputation algorithms have been previously validated and, when checked against our hand-codes, provide reasonable accuracy. NamePrism shows a concordance with our hand-codes of 84% for national/ethnic background and 92% for regional background. While Lauderdale’s accuracy against our hand-codes was lower—at 66%—Lauderdale was previously validated using health data. Further, switching the order in which we used Lauderdale vs. NamePrism made little difference for our estimated rates (see our response to Reviewer 3, Comment 2d).

In terms of our estimated rates, uncertainty from having a small number of deaths per group is empirically reflected in the width of our 95% CIs. Despite large confidence intervals for particular groups, we have sufficient statistical efficiency to make conclusions about differences in rates between those groups. This is especially true for our estimates for regional background, where smaller groups were aggregated into larger ones.

The main uncertainty in our estimates is driven by our unidentified cases. This is clearly an issue that needed to be further addressed. However, our data suggest that these unidentified cases likely come from groups we already identify as having high rates, such that, if anything, our conclusions about which groups are at higher risk would likely be strengthened were we able to identify these cases. We have now included a paragraph on this issue in our discussion:

“Our analysis has key limitations. Most importantly, we could not identify the regional (18%) or national/ethnic (33%) background of all decedents, and because not all police violence fatalities are captured in Fatal Encounters/MPV, the presented results underestimate true rates during the study period. These undercounts are likely differential. Native Hawaiians, for example, were almost certainly undercounted due to limitations in media reports (see Methods) and imputation algorithms. Indeed, Lauderdale’s surname list included only names identified as Chinese, Filipino, Japanese, Vietnamese, Korean, or Indian, while NamePrism’s East/South/Southeast Asian or Pacific Islander countries were limited to Cambodia, Myanmar, Thailand, Vietnam, Indonesia, Malaysia, China/Singapore/Hong Kong/Taiwan, Philippines, South Korea, Japan, and Pakistan, Bangladesh, South Asia/Nepal/India/Sri Lanka. Many of the national/ethnic backgrounds excluded from these lists—including not only Native Hawaiians but also Laos, Hmong people, and all countries in the Pacific Islands—were also groups that already appear to experience among the highest rates of fatal police violence across all national/ethnic groups in this study. Thus differential undercounting likely means our core findings on the elevated risks of Pacific Islanders and Southeast Asian Americans whose families originated in areas impacted by US war in Southeast Asia are likely underestimated and more severe than are presented here.”

6. What percentage of the total relevant FE deaths do AAPI constitute? From my calculation/research, it is a small %. Is it worthy of this much investigation?

RESPONSE: These deaths are, to be sure, a very small proportion of all FE deaths. While a small proportion of FE deaths are AAPI people, however, we argue that the total number of FE deaths is not a meaningful denominator. The relevant estimand is risk for specific groups and disproportionality in that risk, which we assess in this paper. 

We find that for certain AAPI groups, these deaths occur at a high rate – as high or higher than the rate for other racial/ethnic groups who make up larger portions of FE deaths. Given that there is also evidence that deaths in police custody can have wide-ranging implications for the health of people of the decedent’s same racial/ethnic group, these low absolute numbers of deaths may constitute major population stressors, in fact especially when there are fewer people of a particular racial/ethnic group in the US. The fact that the Consul General of the Federated States of Micronesia became directly involved in demanding justice and further investigation in one of only 3 cases of a Micronesian person dying reflects the community-wide, even international, import of these deaths. Further, several studies, many using robust quasi-experimental methods, find that the stress of an incident of fatal police violence for people of the same racial group may cause poorer mental health, worse birth outcomes, and declining cardiovascular health, meaning even a small number of these deaths could have implications for millions of people.

Moreover, treating small ethnic groups as negligibly important to science suggests a troubling orientation to the well-being of the communities in these data. A similar comment about a biomedical article focused on treating an exceptionally rare condition, for example, would be understood to be dehumanizing and unhelpful for people with that condition and their families.

We have edited our paper’s discussion section to explain the relevance of our analysis:

“Furthermore, our findings have broad public health importance given potential implications of fatal police violence for the health of people of the decedent’s community and same racial and ethnic group, which may be even more significant in smaller ethnic groups or tight-knit ethnic communities and/or if these deaths involve immigration enforcement agents (19). Recent studies, for example, have indicated that fatal police violence likely has spillover effects beyond those who are killed, impacting the mental health, birth outcomes, and cardiovascular health of community members with the same racial/ethnic background as decedents (20-23).”

---

## [Editor Report · Decision Letter 1]

6 Sep 2022

Disaggregating Asian American and Pacific Islander Risk of Fatal Police Violence

PONE-D-21-36607R1

Dear Dr. Schwartz,

We’re pleased to inform you that your manuscript has been judged scientifically suitable for publication and will be formally accepted for publication once it meets all outstanding technical requirements.

Kind regards,

Jonathan Jackson, Ph.D

Academic Editor

PLOS ONE
---

## [Editor Report · Acceptance letter]

29 Sep 2022

PONE-D-21-36607R1 

Disaggregating Asian American and Pacific Islander Risk of Fatal Police Violence 

Dear Dr. Schwartz:

I'm pleased to inform you that your manuscript has been deemed suitable for publication in PLOS ONE. Congratulations! Your manuscript is now with our production department. 

Kind regards, 

on behalf of

Dr. Jonathan Jackson 

Academic Editor

PLOS ONE